

# A synthetic cell density signal can drive proliferation in chick embryonic tendon cells and tendon cells from a full size rooster can produce high levels of procollagen in cell culture

Richard I. Schwarz[1,2]

[1] Biological Systems and Engineering Division, Lawrence Berkeley National Laboratory, Berkeley, CA, United States
[2] SNZR LLC, Oakland, CA, USA

## ABSTRACT

Cell density signaling drives tendon morphogenesis by regulating both procollagen production and cell proliferation. The signal is composed of a small, highly conserved protein (SNZR P) tightly bound to a tissue-specific, unique lipid (SNZR L). This allows the complex (SNZR PL) to bind to the membrane of the cell and locally diffuse over a radius of ~1 mm. The cell produces low levels of this signal but the binding to the membrane increases with the number of tendon cells in the local environment. In this article SNZR P was produced in *E.coli* and SNZR L was chemically synthesized. The two bind together when heated to 60 °C in the presence of $Ca^{++}$ and $Mg^{++}$ and the synthesized SNZR PL at ng/ml levels can replace serum. Adding SNZR PL to the medium was also tested on primary tendon cells from adult roosters. The older cells were in a maintenance state *in vivo* and in cell culture they proliferate more slowly than embryonic cells. Nevertheless, after reaching a moderately high cell density, they produced high levels of procollagen similar to the embryonic cells. This data was not expected from older cells but suggests that adult tendon cells can regenerate the tissue after injury when given the correct signals.

## INTRODUCTION

Cell density signaling is fundamental to tendon morphogenesis but this is not obvious until one understands both the developmental and cellular requirements. A tendon is basically a collagen rope that connects a muscle to a bone and needs to be precisely the right length and the right diameter to match the size and weight of the organism at every stage of development. Because of these requirements, tendon cells need to coordinate collagen production and cell proliferation to the growth of the whole organism. The embryonic chick tendon cells have to make 60% of their total protein production as procollagen and proliferate rapidly (*Dehm & Prockop, 1971*) while the adult tendon cells in a maintenance state only make 1% procollagen and grow very slowly (*Schwarz, Farson & Bissell, 1979*).

Corresponding author
Richard I. Schwarz, rischwarz@lbl.gov

Tendon cells transition from being one of the most differentiated cells in the body to one of the least differentiated over the course of development. Nevertheless, it is considered to be one of the simplest tissues in the body because it is almost all one cell type and one protein, ~90% collagen (*Amiel et al., 1984*).

The developmental constraints in making a perfect collagen rope at every stage in development make it difficult to visualize how this could be regulated at the cellular level. The assumption that a tendon cell should be a simple perturbation of the typical cell neglects the extremes that the tendon cell needs to accomplish. When chick tendon cells are grown in low serum medium (0.2%) with ascorbate, one observes that the high procollagen production drops quickly to ~12% when the cells are seeded at low cell density and then returns to around 60% procollagen as the cells grow to high cell density. How cell density regulates the procollagen pathway at a post-translational step is not required for this article and it has been recently reviewed (*Schwarz, 2015*). The important point is that procollagen production is an excellent indicator of tendon cell density.

In cell culture, one can manipulate tendon cells and their environment to reveal important characteristics of cell density signaling. In previous experiments, the most useful variation was seeding cells inside a 6 mm cloning ring that was inside a 60 mm cell culture dish. Once the cells had attached to the dish, the cloning ring was removed, 5 ml of medium was added and the cells were allowed to grow. The cells formed an island where they were at high cell density in the middle of the island and then gradually lower cell density as it approached the edges of the island. *In situ* hybridization for procollagen mRNA revealed an increasing gradient of procollagen mRNA with increasing density of cells over a distance of ~1 mm (*Schwarz, 1991*). This seemed more consistent with a short-range diffusible factor than with increasing cell-to-cell contact. To further test this, cells grown as islands were gently rocked in the incubator. These cells after 1 day lost their ability to sense cell density. Cells that were confluent in the middle of the island started to grow with a 48 h generation time while the level of procollagen mRNA dropped to one third of the level seen in the unshaken controls. This was consistent with the concept that the cell density signal was a short-range diffusible signal. Because shaking into a large volume of medium caused the cells to grow, the factor appeared to be a growth inhibitor. To test this, shaken conditioned medium was added to cells growing in an island and the result was that this factor stimulated growth even in the cells at high cell density (*Schwarz, 1991*). This is a conundrum: adding or removing this factor caused cells to grow. This is usually an indication that the answer has been oversimplified. The solution to this problem is to add a second factor. In this case, the second factor is not diffusible, has no activity on its own, but interacts with the diffusible factor and the interaction between them inhibits proliferation and stimulates procollagen synthesis.

Mathematical transformation of this two-factor model by integration changes rates of cell proliferation and rates of procollagen production and into a pattern formation of where the cells are distributed and collagen fibrils are laid down. In earlier studies, a computer was used to solve this problem and generate a simulation of tendon morphogenesis (*Schwarz, 1996*). This correlates extremely well with the dramatic changes seen in tendon tissue as it matures from an embryonic to an adolescent state (Fig. S1).

The computer simulation also predicts that tendon elongation is driven by a growth plate where cells at the front at moderate cell density are growing, where cells in the middle at high cell density are making high levels of procollagen, and where cells at the back of the growth plate are apoptosing (*Petzold & Schwarz, 2013*; *Schwarz, 1996*). This explains how tendon cells can lay down a uniform "rope" of collagen that precisely fills the gap between the bone and the muscle as the organism grows during development (*Petzold & Schwarz, 2013*). The growth plate at the tendon\muscle junction has been confirmed by histological sections of adolescent chickens, and nascent humans (*Williams, 1995*; *Schwarz, 1996*).

To further confirm this finding we purified, in an earlier set of experiments, 1 µg of the first factor from 400 l of shaken conditioned medium in collaboration with scientists at Amgen. This turned out to be a composite molecule composed of a small protein, 94 AA (amino acid) protein (SNZR P), highly conserved between chicken and human, bound tightly to a tissue-specific lipid (668.6 MW; SNZR L) (*Petzold & Schwarz, 2013*). The first factor is diffusible because of the small protein and the activity is predicted to be due to the unique, tissue-specific lipid. Because of the composition of the first factor, the second factor is hypothesized to be the free lipid (*Schwarz, 2015*).

In this article we are making the final confirmation that we have the correct molecule for SNZR PL by making a completely synthetic form and testing its ability to stimulate chick tendon cells to proliferate. In the second part of this article, primary tendon cells from adult roosters were grown in culture and tested to see if they remain responsive to ascorbate induction of procollagen synthesis.

## MATERIALS AND METHODS

### Cells and cell culture

Embryonic tendon cells were isolated from 16 day chick embryos by a modification *Petzold & Schwarz (2013)* of the procedures described by *Dehm & Prockop (1971)*. The basic medium used was a 50/50 mix of F12/DMEM and if serum was used it was fetal bovine serum deactivated at 56 °C for 30 m.

Adult tendon cells were from full size roosters that had been freshly killed at the local Muslim butcher. The rooster was washed in a disinfecting solution, ~1% solution of BDD (Fisher Scientific, Waltham, MA, USA) for ~5 min and then ~3 min wash in 70% ethanol. The legs were removed. Then the shank dissected from the foot to the hock. The thick, scaly, yellow skin covering the shank was slit open revealing a sheath with several tendons running through it. The tendons were dissected and removed from the sheath, cut into small pieces, and then put in dissociation medium (20 ml) containing 120 mg collagenase (type I; Sigma or Worthington, Columbus, OH, USA); 80 µl bovine serum albumin (at 1 mg/ml; Sigma, Columbus, OH, USA); 200 µl, 100x penicillin/streptomycin. The 50 ml centrifuge tube was incubated horizontally for 3 h at 37 °C on a rotator at 100 rpm and for the last 20 m, 0.5 ml of 1 mg/ml of DNAse (Cooper Biomedical, San Ramon, CA, USA) was added to the dissociation solution. Then, the tube was held upright and the large undigested tendon tissue was allowed to settle to the bottom. The supernatant was collected and 10ml of medium was added and tube inverted several times. Again the second supernatant was collect and added to the first. This was put through a nylon mesh

(192 μ) to further remove large fragments. At this point the cells were spun down in a small clinical centrifuge (speed 6 out of 7), resuspended in 20 ml of fresh medium and counted on a hemocytometer slide.

The 16 day chick embryos were exempt from requiring an animal use protocol as determined by the LBNL Animal Welfare and Research Committee. Similarly, the purchase of a dead rooster from the butcher shop was also an exempt use.

## Freezing embryonic tendon cells freshly isolated from the tendon

Tendon cells were spun down and resuspended in freezing media, 90% FBS and 10% DMSO, at $4 \times 10^6$ cells/ml. A total of 1 ml aliqouts were put into a CoolCell (Corning, Corning, NY, USA) and this was put into a −80 °C freezer. Within a couple days the tubes were transferred to a liquid nitrogen freezer. When needed, cells were quickly thawed and 250 μl of cells were added to a flask containing 10 ml of F12/DMEM medium. The cells were placed in a 5% $CO_2$ incubator for 3 h and then the medium was changed to the appropriate growth medium.

## Production and purification of SNZR P

To make an exact copy of the 94 AA protein (SNZR P), an Impact kit (New England BioLabs, Ipswich, MA, USA) was used with the following modifications. The vector pTYB21 was used so that an N-terminus methionine would not be added to SNZR P. Because the cDNA encoding the 94 AA protein contains Sapl restriction enzyme site, Sapl could not be used in the cloning. Instead, another kit was used, NEBuilder HiFi DNA Assembly Master Mix/NEBuilder HiFi DNA Assembly Cloning (New England BioLabs, Ipswich, MA, USA) that allows assembly of the vector to SNZR P without the need for restriction enzymes. Another change was to transform the *E. coli* strain DH5a first and then subcloned into T7 express (New England BioLabs, Ipswich, MA, USA). The induction of these cells used 0.4 mM IPTG at 30 °C for 5 h. The sonicator (diagenode, Denville, NJ, USA) used to break open the *E. coli* was set at 4 °C at high power with 20 cycles of 30 s on/30 s off. The other directions in the kits were closely followed.

## SDS gel electrophoresis

The standard gel format was Novex™ 4–20% Tris-Glycine Mini Gels, WedgeWell™ format, 10-well, 1 mm thick (Invitrogen, Waltham, MA, USA) run for 90 m at 120 v. Samples were diluted with equal amounts of 2x sample buffer (Invitrogen, Waltham, MA, USA), β-mercaptoethanol was added (5%),and then heated at 80 °C for 10 m. In the case where we were tracking the production of SNZR P there was sufficient amounts of protein that the gels were stained with Simply Blue (Invitrogen, Waltham, MA, USA) and photographed with a standard camera and light box. When we were tracking the loss of the SNZR P band when it would bind to SNZR L, the levels were in the ng range so the staining shifted to Sypro ruby (Invitrogen, Waltham, MA, USA) and bromphenol blue was removed from the sample buffer in order to reduce background fluorescence at the front of the gel. Because of limited access to the lab during COVID-19, the Sigma protocol was used where the gel was run and fixed on 1 day and stained on another day. The gels were

**Table 1 Calculations from raw data to corrected ng procollagen.**

| Day | Ascorbate, SNZR level | Band intensity procollagen1 α1 | Band intensity procollagen1 α2 | Calculated ng/band procollagen1 α1 | Calculated ng/band procollagen1 α2 | Total ng procollagen | Ratio α1/α2 | Serum band intensity | Average serum band intensity | Correction factor | Corrected ng procollagen |
|---|---|---|---|---|---|---|---|---|---|---|---|
| 17 | +C, 0x | 1153558 | 695062 | 14.3 | 8.6 | 22.9 | 1.7 | 4250469 | 4897535 | 0.87 | 26.4 |
| 17 | +C, 3x | 1209985 | 769643 | 15.0 | 9.5 | 24.5 | 1.6 | 5013557 | | 1.02 | 24.0 |
| 17 | +C, 5x | 1441538 | 906306 | 17.9 | 11.2 | 29.1 | 1.6 | 5513915 | | 1.13 | 25.9 |
| 17 | +C, 10x | 1136161 | 770680 | 14.1 | 9.6 | 23.6 | 1.5 | 4812199 | | 0.98 | 24.1 |
| 18 | +C, 0x | 2701950 | 1273890 | 33.5 | 15.8 | 49.3 | 2.1 | 6223518 | 6126505 | 1.02 | 48.5 |
| 18 | +C, 3x | 1717248 | 1078740 | 21.3 | 13.4 | 34.7 | 1.6 | 6601084 | | 1.08 | 32.2 |
| 18 | +C, 5x | 1339660 | 792978 | 16.6 | 9.8 | 26.4 | 1.7 | 6119600 | | 1.00 | 26.5 |
| 18 | +C, 10x | 403494 | 596042 | – | – | – | – | 5561818 | | 0.91 | – |
| 19 | +C, 0x | 2722274 | 1346254 | 33.8 | 16.7 | 50.4 | 2.0 | 5379358 | 4859917 | 1.11 | 45.6 |
| 19 | +C, 3x | 1981642 | 1198200 | 24.6 | 14.9 | 39.4 | 1.7 | 4620722 | | 0.95 | 41.5 |
| 19 | +C, 5x | 1801835 | 1144047 | 22.3 | 14.2 | 36.5 | 1.6 | 5730756 | | 1.18 | 31.0 |
| 19 | +C, 10x | – | – | – | – | – | – | 3708832 | | 0.76 | – |
| 20 | +C, 0x | 3977670 | 1924041 | 49.3 | 23.9 | 73.2 | 2.1 | 4843214 | 3818935 | 1.27 | 57.7 |
| 20 | +C, 3x | 2003493 | 774109 | 24.8 | 9.6 | 34.4 | 2.6 | 3850674 | | 1.01 | 34.2 |
| 20 | +C, 5x | 2283199 | 1276692 | 28.3 | 15.8 | 44.1 | 1.8 | 2900423 | | 0.76 | 58.1 |
| 20 | +C, 10x | 1377030 | 698967 | 17.1 | 8.7 | 25.7 | 2.0 | 3681427 | | 0.96 | 26.7 |

**Note:**

The raw data from the imager for the intensity of the procollagen bands was converted to ng using the standard curve (Fig. S2). From this data the ratio of the bands (α1/α2) was calculated. A correction was also made for variations in loading and staining. For this, the serum band running at 169 kD was used as the control. For each day the average intensity of this band was calculated and then used to make a correction factor so that the intensity of that band would be uniform for the four lanes (+C, 0x; +C, 3x; +C, 5x; +C, 10x) . This correction factor was then applied to procollagen bands to yield a corrected ng procollagen.

photographed using excitation at 472 nm and an emission at 684 nm (Azure Biosystems 600; Azure Biosystems, Dublin, CA, USA). The Azure Biosystems software was used to quantitate the bands. The recommended rolling ball method was used to define background. A small value for the ball (#4) allowed the software to draw an accurate boundary between the fluorescent signal and the background. When using RNase A (Worthington, Columbus, OH, USA) to make a standard curve between 3 and 21 ng (Fig. S2), the software generated a linear fit going through the origin. Similarly, the software quantitatively calculated the ratio of the procollagen α1 band to the procollagen α2 band as 1.8 ± 0.3, $n = 14$ (Table 1).

# RESULTS

## Chemical synthesis of SNZR L, the unique lipid of the chicken tendon cell density signal

In a previous article (*Petzold & Schwarz, 2013*) a working model of SNZR L was developed and is now shown as the "original" lipid structure (Fig. 1). This model was consistent with mass spectrometry (ms) data and enzymatic data on specific lipid cleavage sites. Other

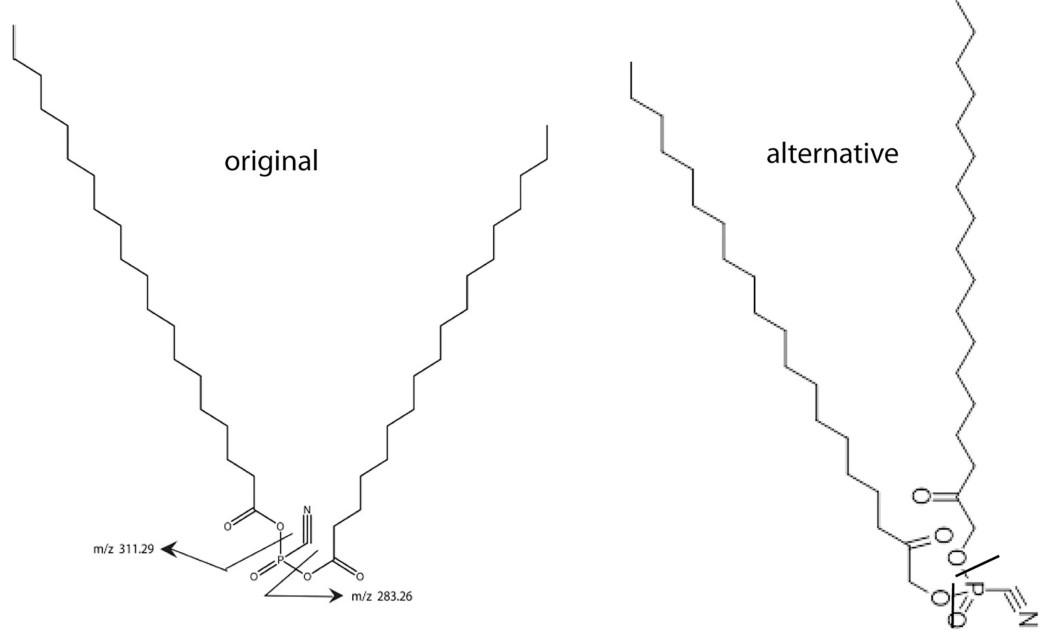

**Figure 1 Two models for SNZR L.** Two models for SNZR L both with the same molecular weight, the same predicted ms/ms fragmentation pattern but with different bonding to the phosphate. The "original" model joined two fatty acids to a phosphate using two high energy, acid anhydride bonds. Scientists at Avanti Polar Lipids suggested an "alternative" model where the fatty acids were reduced, a common metabolite, and then the joining would be a stable ester bond. Previous research had shown that SNZR PL was a very stable molecule, so the "alternative" became the preferred model.

analytical methods were not available because they required a higher concentration of the lipid than we could produce. Our goal was to synthesize a biological active lipid knowing that a small error in the structure usually results in little or no activity. Our only solution for confirming the lipid structure was to synthesize the structure that best fit all the available data and then test its biological activity on tendon cells.

This came into focus when Avanti Polar Lipids was contracted to make the original lipid. They realized that joining a fatty acid to a phosphate, forms a high energy, acid anhydride bond. This could be problematic because the intermediate form might not be stable enough to allow a second fatty to be added. The synthesis might require that both fatty acids be added at the same time resulting in a mixture of three products. And even then, these molecules might not be stable. A group of lipid chemists and ms specialists at Avanti Polar Lipids met and discussed this problem. They came up with an "alternative" structure (Fig. 1) where the fatty acids are reduced yielding an alcohol with a ketone, a common metabolite. This would change the bond to a stable ester while not altering the MW of the molecule or the ms/ms fragmentation pattern.

The better candidate between the "original" form and the "alternative" forms relied on our characterization of the active molecule, SNZR PL. Previous experiments with SNZR PL (in conditioned media) showed that it was stable to heat, 90 °C for 10 min (*Zayas & Schwarz, 1992*), to storage at 4 °C in conditioned medium for 1 year, and to purification through four columns with changes in salt concentration, pH, and solvents

(*Petzold & Schwarz, 2013*). Stability strongly favored the "alternative" structure. Avanti Polar Lipids synthesized 0.4 g from which a concentrated solution was made by adding 0.1 mg to 50 ml of ethanol (95%) and this solution has been stable for over a year at 4 °C.

## Molecular synthesis of the 94 amino acid (AA) chicken protein, SNZR P

SNZR P is the opposite of SNZR L being highly conserved between species and tissues. Even the larger gene, 424 AA from which SNZR P is cleaved at the carboxy end, is highly conserved between chicken and human (*Petzold & Schwarz, 2013*). The chicken form can be an active participant in human cells even with the addition of a methionine at the amino terminus and tags at the carboxy terminus. Nevertheless, in making a synthetic version for the first time it was important to make the protein identical to the native form to reduce the chance for potential artifacts. Because SNZR P was shown to be unmodified when expressed in human and chicken cells (*Petzold & Schwarz, 2013*), this protein can be made in *E. coli*.

The IMPACT kit from New England BioLabs was developed in part for this purpose. The *E. coli* expression system makes a 56 kD protein that can be extended with our 94 AA protein (10.8 kD) yielding a composite protein of 66.8 kD. The 56 kD portion has two important functions. It has a chitin binding site that allows it to bind tightly to a chitin column. This allows the separation the 66.8 kD protein from the vast majority of *E. coli* proteins. The second property is inclusion of an intein, an enzyme that cleaves at its carboxy terminus when induced with a reducing agent. This releases the 10.8 kD protein in almost pure form (Fig. 2). A liter culture of *E. coli* yields approximately 1 mg of SNZR P.

## Reconstituting SNZR PL from its component parts

SNZR P and SNZR L are bound tightly together in the active signaling molecule. Since the protein can have regions where it is hydrophobic and other regions where it is hydrophilic, these could interact independently to bind to similar regions in the phospholipid. There are also divalent ion interactions because chelators are known to inhibit activity (*Zayas & Schwarz, 1992*).

The binding will change its apparent molecular weight on SDS gels from ~11 to ~16 kD and activate its biological activity. The protein by itself has no biological activity (*Petzold & Schwarz, 2013*) and the lipid biological activity by itself is not known. The first two attempts to recombine SNZR PL from its components focused on biological activity. The first was to mix equal molar amounts SNZR P with SNZR L in PBS at room temperature at 1,000x, where x = 100 pg/ml of SNZR P. The second attempt was mixing equal molar amounts with 0.5 mM $Ca^{++}$ and 0.5 mM $Mg^{++}$ at 37 °C for 17 h at 10,000x. In these two cases, testing heat, concentration, and divalent ions for their ability to drive the two components together and form an active molecule. Initially, when added to primary cultures of embryonic chick tendon cells, all the concentration allowed the cells to attach to the flask and seemed to be growing (1x, 5x, 10x, 25x, and a control with 0.2% FBS (fetal bovine serum)) but by the second medium change on the 4th day, cells given high levels stopped growing and all of them were not confluent by the 7th day in sharp contrast to the control cells growing in low levels of FBS. Simply stated, the data from this

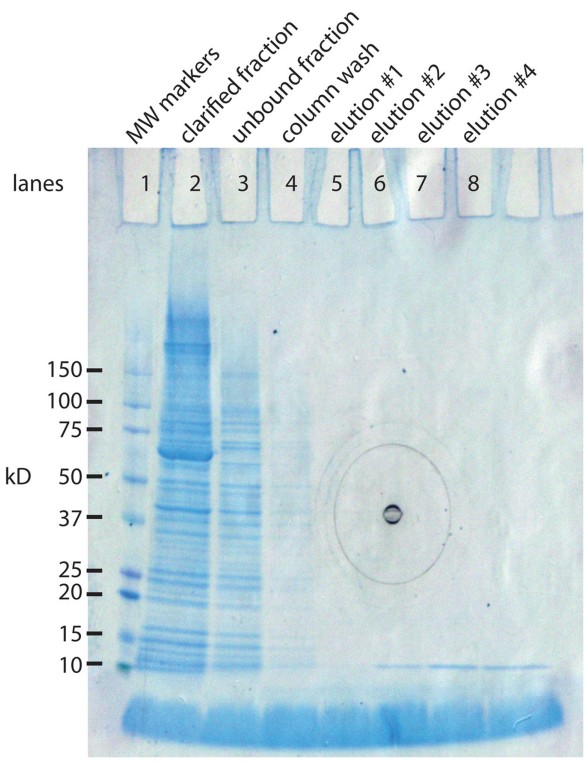

**Figure 2  SDS gel analysis of the purification steps in the synthesis of SNZR P.** The 94 AA protein is made in *E. coli* as a fusion protein. The vector, pTYB21 (New England BioLabs, Ipswich, MA, USA), is a T7 promotor driven system with a chitin binding site so that the fusion protein can bind to chitin column for puification. The fusion protein also contains an intein, a self-cleaving enzyme, that cleaves at its carboxy terminus when treated with a reducing agent. By attaching the 94 AA protein to the carboxy terminus of the intein, it can be released from the chitin column simply by adding a reducing agent. By sampling various steps, the synthesis of the fusion protein, the binding to the chitin column, and the release of the 94 AA protein can be followed on a SDS gel stained with coomassie blue. Lane 1: molecular weight markers. Lane 2: clarified fraction: *E. coli* induced expression for 5 h, sonicated, and a sampling of the soluble proteins. The most intense band in the lane is at 66.8 kD, the expected size of the fusion protein. Lane 3: unbound fraction: sampling of the proteins that did not bind to the chitin column. Lane 4: column wash. Lane 5–8: elution #1–#4. a reducing agent was added to the column and 48 h latter eluted in 2 ml fractions from the gel. The 10.8 kD band is the expected molecular weight for the 94 AA protein. Elutions #5–#8 were run on a separate gel, data not shown (the black concentric circles in the middle of the gel are manufacturing marks on the outside gel container).

experiment was unexpected. Furthermore, using a biological assay when the role of SNZR L was not better understood only added to the complexity of the problem.

To gain clarity, our focus shifted to gel analysis of the assembly of SNZR PL from its two components. Preliminary experiments showed that Sypro ruby, a red fluorescent protein dye, had sufficient sensitivity to detect ng levels of SNZR P. These experiments also showed that heating for 10 min at 50 °C or 60 °C in the presence of divalent ions would reduce the SNZR P signal while 37 °C was not sufficient. These experiments also showed that we could not detect the SNZR PL band. This is expected because adding a lipid component to the protein could alter the binding of the protein dye. A similar problem had been seen before with Western blotting when different monoclonal antibodies to SNZR P, or to tags

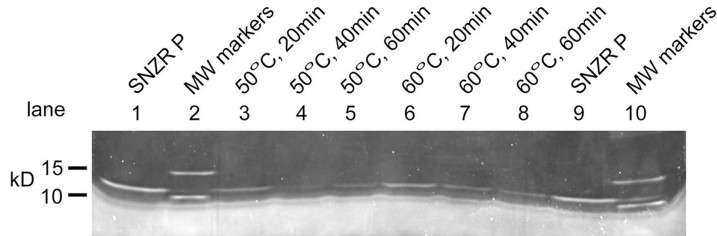

**Figure 3 Reconstituting SNZR PL, gel analysis.** Reconstituting SNZR PL from its component parts requires incubation at elevated temperature (50 °C or 60 °C) for 1 h. The assay looks at the loss of the SNZR P band at 10.8 kD over time on a SDS gel stained with Sypro ruby (uncropped gel Fig. S4). One would also expect to see the appearance of a new band for SNZR PL, however, a small protein bound to a phospholipid seems to resist this stain. Lanes 1 and 9: SNZR P only. Lanes 2 and 10: molecular weight markers. Lanes 3–5: incubated at 50 °C for 20, 40, and 60 min respectively. Lanes 6–8: incubated at 60 °C for 20, 40, and 60 min respectively. The similar kinetics and high loss of the SNZR P band at both temperatures indicated that either temperature was effective.

added to SNZR P, would not bind well to SNZR PL (*Petzold & Schwarz, 2013*). During a one hour time course experiment at both 50 °C and 60 °C, the SNZR P band is reduced by about two third by 20 min and ~90% by 60 min using either temperature (Fig. 3). The ~10% of SNZR P that does not find its SNZR L partner, implying there is a small excess, is the preferred result because SNZR P alone does not have biological activity. Because the transition between working at 50 °C and not working at 37 °C has not been defined, 60 °C would offer a safer margin for our standard conditions for recombining SNZR PL.

## Biological activity of reconstituted SNZR PL

To confirm that the reconstituted SNZR PL were correctly aligned and biologically active, it was tested on embryonic tendon cells at 1x, 5x, and 10x and a control with 0.2% FBS. In contrast to the earlier experiment, cells continued to grow over the week but did not become confluent except for the control cells. This data triggered a standard response when cells do not grow fast enough—add more factor. So we tested 0x, 5x,10x, 20x, 40x, 80x, 0.2% FBS on primary avian tendon cells. In basal medium, cells seeded poorly and grew slowly. Initially, the higher levels (≥10x) would grow better than the 5x cultures but growth would slow by mid week. This is the expected behavior in our two factor model where feedback control is important part of the model. In the next experiment, the focus was on 5x cultures and growing them for longer times. Nevertheless, after 9 days they had grown well but were not confluent. At that point, we drove them to confluency by raising the level of SNZR PL to 10x for 2 days (Fig. 4, day 11).

The simple conclusion is that one can reduce the protein level in 0.2% serum a thousand fold by using SNZR PL and stimulate primary avian tendon cells to grow from low cell density to a confluent culture. This is a good confirmation that the reconstituted SNZR PL are biologically active. But this result helps shed light on several other issues. For instance, why does growing embryonic tendon cells with 0.2% FBS allow the cells to grow faster and reach confluent densities in 6 days instead of the 11 days using SNZR PL? This very low

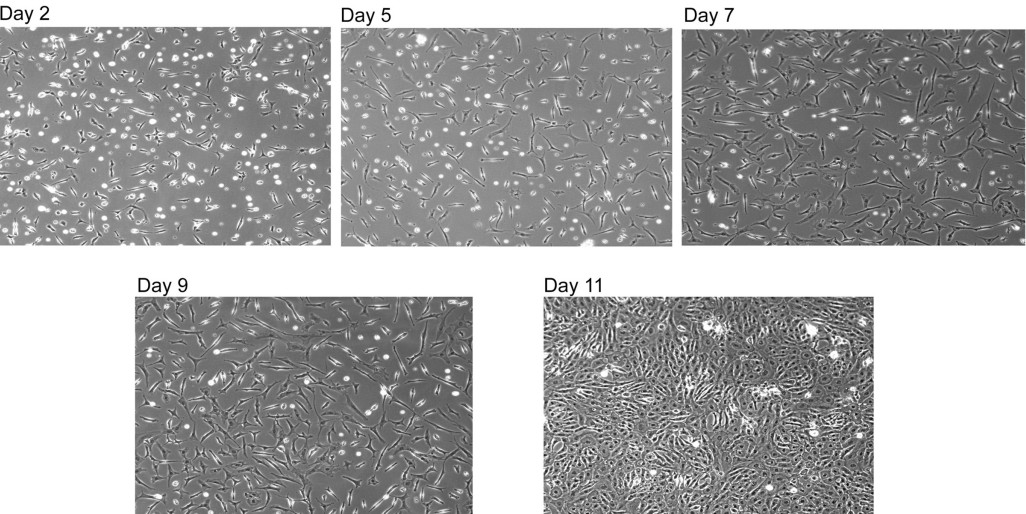

**Figure 4 Reconstituted SNZR PL drives cell proliferation.** A reconstituted SNZR PL from component parts can drive the proliferation of embryonic tendon cells. Frozen cells were thawed and $1 \times 10^6$ cells were seeded into a 25 cm² flask with 10 ml of F12/DMEM medium and incubated for 3 h to allow the cells to attach. The medium was changed to growth medium, F12/DMEM + 5X SNZR PL (0.5 ng/ml). Day 2: seeding efficiency ~50% and a lot of rounded up dead cells can be seen. Day 5: fewer dead cells with proliferation of the live cells. Day 7 and Day 9: continued proliferation. Day 11: because the cells on day 9 had not become confluent, the level of SNZR PL was doubled to 10X and this drove proliferation of the cells to become confluent.

level of serum probably does not influence the cell as a growth factor when serum usually is used at 25 to 50 fold higher levels. Low levels of serum are more likely to stimulate the metabolism of the cell (*Valmassoi & Schwarz, 1988*), and as a consequence, the cell produces more SNZR PL. This would raise the basal level but at a local level and at lower amounts. In comparison to standard cell culture approach where we add ng/ml of SNZR PL to the medium, flooding the environment. This can quickly overload the response triggering feedback control. So having a good delivery system for SNZR PL is important in producing a desired response from the cell.

These experiments also give us insight into SNZR L when not bound to its protein partner. In the early experiments to combine SNZR L with SNZR P where the temperature was 37 °C, no binding occurred and the biological activity was due to SNZR L. Free SNZR L outside the cell, even at low levels, acts like high levels of SNZR PL. Free SNZR L would still interact with the membrane of the cell but would no longer be diffusible, effectively increasing its biological activity. Increasing its levels with each media change, it would quickly transition from a stimulator of growth to an inhibitor of growth. Nevertheless, free SNZR L is not normally found outside the cell and its role inside the cell should remain as predicted in our model. Only now SNZR L binds to the inside of the plasma membrane and this can influence SNZR PL that binds to the outside of the membrane. When SNZR L on the inside membrane is close enough to interact with SNZR PL, this "complex" changes the biological signaling to an inhibitor of proliferation and a stimulator of procollagen production. While SNZR L continues to be a good candidate for the second factor, this remains to be determined.

## Proliferation and differentiation in adult chick tendon cells

Embryonic chick tendons are formed rapidly so that the hatched chick can walk and find food and water. The length is controlled by a growth plate that is postulated to interact with muscle connective tissue to insure that the tendon is always the correct length as the organism grows (*Petzold & Schwarz, 2013*). The growth plate "moves" by having cells at the front proliferate, cells in the middle differentiate, and cells at the back apoptose. A small subset of cells at the periphery of the growth plate never reaches high cell density and do not apoptose. These cells end up between collagen fibrils where they make the collagen that expands the diameter of the fibril so that the tendon is strong enough to support the weight of the growing chicken (Fig. S1). When full size is reached, tendon cells need to reduce proliferation and collagen production to maintenance levels. How is this transition achieved and is it reversible, at least to some extent, by changes in the levels of SNZR PL?

Releasing cells from the tendons of full size roosters is a challenge. The tissue is made up of highly dense, cross-linked collagen with relatively few cells residing between the fibrils (Fig. S1). To free the cells one has to incubate the tissue for several hours in collagenase. This lowers the viability of the released cells. In this experiment, the cells were seeded at $36 \times 10^3$ cells/cm$^2$ but the number of viable cells was estimated at only 25%. This lower cell density slows initial growth and pushes the cells to grow in patches because cell distribution is not uniform, cells in moderate density areas grow better than cells in low density areas. This low viability also restricts our ability to analyze growth curves. So our focus was on how adult tendon cells respond to cell density and ascorbate in their regulation of procollagen production.

Cells were grown in medium with 0x, 3x, 5x, or 10x SNZR PL. Cells grew slowly and on the 12$^{th}$ day the medium was changed to include 0.2% serum to all conditions. This increased cell growth and on the 16$^{th}$ day, half the flasks were given ascorbate (50 μg/ml) and the medium changed daily. The conditioned medium (24 h) was collected and saved over the next 4 days to assay for collagen production. Embryonic chick tendon cells "synthesize and secrete prodigious amounts of procollagen (about 0.6 mg/$10^9$ cells/h) for the first 8 h or so in culture" (*Prockop, Sieron & Li, 1998*). One can calculate that in embryonic cells induced with ascorbate—10 ml/flask, ~$10^6$ cells, incubated for 24 h—that taking 35 μl and loading it on an SDS gel would yield ~30 ng of a procollagen α1 band and ~15 ng of a procollagen α2. This would be easily detected using Sypro ruby staining. However, it would be just faintly detectable in uninduced cells (minus ascorbate) making 6-fold less procollagen. In earlier experiments, adult tendon tissue made ~1% (*Schwarz, Farson & Bissell, 1979*) and this would be undetectable. Using this assay, the conditioned media from adult tendon cells on days 17, 18, 19, and 20 were assayed for procollagen production. The Sypro ruby stained gel with samples from day 20 is shown in Fig. 5. As expected, only serum proteins are prominent in all lanes (compare lane 10, medium only). Since we are only applying 1/286 of the total secreted proteins to the gel, this is too little to detect except for procollagen bands with a molecular weight around 150 kD, especially in the ascorbate (+C) induced cells (lanes 6–9) and faint bands in the uninduced cells (lanes 2–5). The upper band is present at twice the level of the lower band (1.8 ± 0.3,

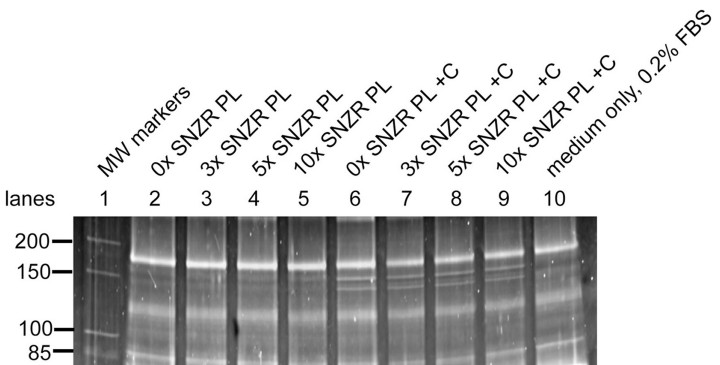

**Figure 5 Procollagen production in adult rooster tendon cells in culture.** Tendon cells from an adult rooster are in a maintenance state of low proliferation and low procollagen production but this is a reversible state. In cell culture these adult cells can be grown to high cell density (day 20) enabling them to make high levels of procollagen. This figure shows a portion of a SDS gel that has been stained with Sypro ruby (the uncropped gel is shown in Fig. S3). Lane 1: Molecular weight markers. Lane 10: 35 μl sample containing only medium with 0.2% FBS. For the remaining lanes 2–9, this is the assay for procollagen produced and secreted into the medium (10 ml) on day 20 (24 h incubation) and a 35 μl sample applied to the well (1/286 of the total). Lanes 2–5: medium + 0.2% FBS containing 0x, 3x, 5x, and 10x of SNZR PL, respectively. Lanes 6–9: medium + 0.2% FBS + ascorbate (+C) containing 0x, 3x, 5x, and 10x of SNZR PL, respectively. The bands not present in the medium only control (lane 10) had a molecular weight of 153 and 140 kD, were highly inducible with ascorbate, and the higher band was two fold more intense than the lower band. These are all characteristics of procollagen α1 and α2 proteins. We should point out that we used the serum band at the top of the gel (166 kD) as a control for two problems. One, the staining of the gel was stronger for lanes 2–6 than for lanes 7–10. A correction factor was determined for each lane so that the intensity of the serum protein was equivalent and this was applied to the procollagen bands as well. Second, there is cupping of the bands, lane 5 in the middle of the gel runs slightly faster than lane 2 at the edge of the gel. By using lane 2 to determine the molecular weight the serum band (being closest to the markers in lane 1), a correction factor was determined so that all the lanes gave the serum band the same molecular weight. This correction factor was applied to the procollagen bands (for details see Table 1 and Table S1).               

$n = 14$; Table 1). The molecular weight of the upper band was 152.7 kD ($\pm 2.2$, $n = 11$) and the lower band 139.8 kD ($\pm 2.9$, $n = 11$). The curvature of the bands on the gel from the outside lanes to the middle was corrected by calculating the molecular weight of the bright serum band (169.1 kD) running in a lane closest to the MW markers. In other lanes, changes in apparent molecular weight of that band because of curvature would be calculated and a correction applied to other bands in that lane. This only altered the average molecular weight of the procollagen band by a small amount but it reduced the standard deviation by 80% (Table S1). These three parameters were all consistent with the upper band being procollagen α1 and the lower band being procollagen α2.

Quantifying the bands intensity was done on Azure Biosystems 600 imager. Sypro ruby is known to give a linear increase in fluorescence with protein concentration. RNAse A was used to generate a standard curve (Fig. S2) and this was used to translate band intensity to ng/band. A correction was also made for loading and/or staining variations. For instance, in Fig. 5 the intensity of the serum band running at 169 kD is higher on the left side (lanes 2–6) than on the right side (lanes 7–10). We took the average intensity of this serum band in the +C bands and used that to create a correction factor for each lane (Table 1). We should point out a limitation of this assay is a lack of internal control. When one

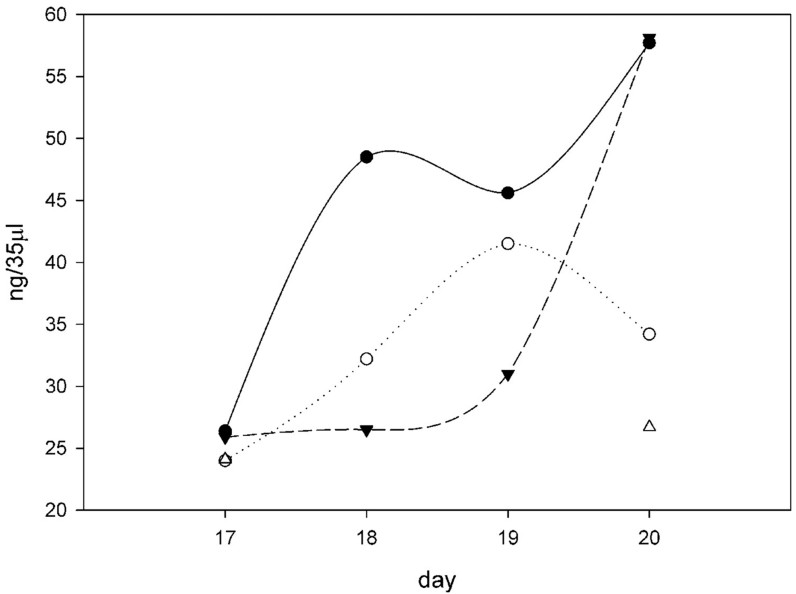

**Figure 6 Procollagen production after ascorbate induction.** Plotting the ng of procollagen in 35 μl of conditioned medium (10 ml total, 24 h incubation period) in cultures induced with ascorbate in different growth conditions over the course of 4 days (days 17–20 from the time of initial seeding of the cells). All the flasks contained 10 ml F12/DMEM + 0.2% FBS + 50 μg/ml ascorbate and then different amounts of SNZR PL, 0x (●), 3x (○), 5x (▲), and 10x (△). The gels, an example on day 20 is shown in Fig. 5, were quantitatively analyzed using software that came with the imager (Azure 600; Azure Biosystems, Dublin, CA, USA; Table 1). The intensity of the fluorescence was converted to ng using a standard curve using RNase A as the standard (Fig. S2). All the points cluster around 25 ng on day 17. This is due in part because ascorbate had been given 24 h earlier and full induction requires 48 h. For 10x, only days 17 and 20 had sufficient levels to quantitate. The shape of the curves is less important for this analysis than the fact that the average level for three conditions (0x, 3x, and 5x) over 3 days (18, 19, 20) is 42 ± 12, a high level. The minus ascorbate levels would be six fold lower, a moderate levels, and be just detectable visually. A maintenance levels at 1% would be undetectable using this assay.

measures percent procollagen production, the non-collagen protein synthesis compensates for changes that could affect all proteins. For instance, if the cells run out of nutrients during the 24 h incubation, this would reduce all protein synthesis. Similarly, an under estimation of cell number would over estimate procollagen production. However, in this case, the induction of procollagen synthesis by ascorbate and cell density can be as much as 60-fold and this becomes the overwhelming change that we are observing. Another advantage is that this procollagen assay has minimal manipulation.

A plot of days after ascorbate induction *vs* ng procollagen synthesized per 35 μl sample was created (Fig. 6). On day 17 all samples were clustered around 25 ng procollagen and this is probably due to a restriction that ascorbate induction requires two days to reach maximum levels. The samples taken from the 10x SNZR PL+0.2% FBS+C made lower levels and only on days 17 and 20 were the levels sufficient to be analyzed. The other conditions 0x SNZR PL+0.2% FBS+C, 3x SNZR PL+0.2% FBS+C, 5x SNZR PL+0.2% FBS +C were all making high levels of procollagen on days 18, 19, and 20. If the three conditions and nine independent assays are combined, the average value 42 ± 12 ng of procollagen strongly supports that adult tendon are capable of making prodigious levels of

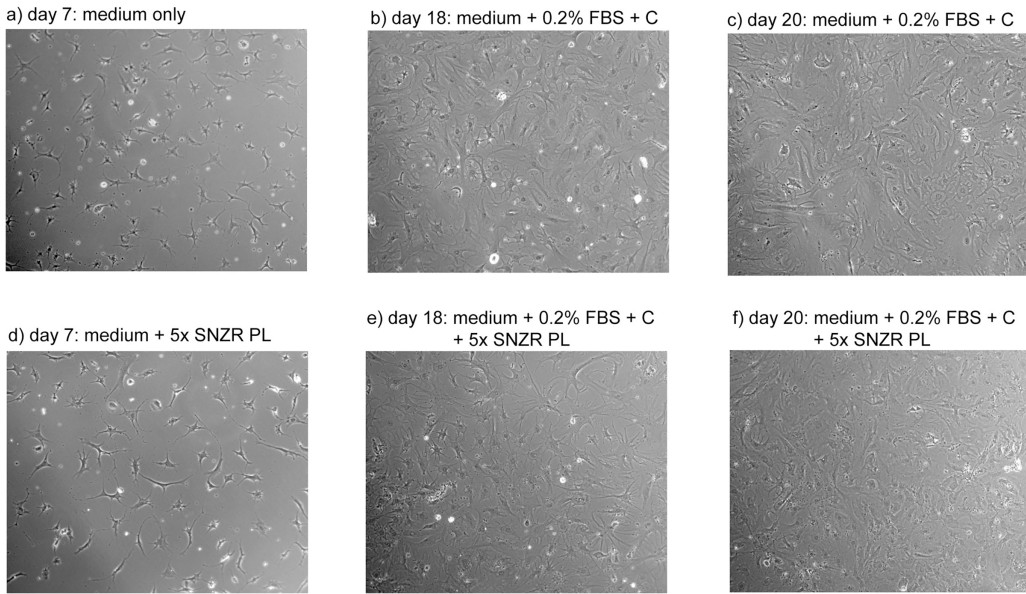

**Figure 7 Proliferation of adult rooster cells in cell culture.** Adult tendon cells growing in cell culture were photographed on days 7, 18, and 20. Shown in the figure are the two conditions that expressed the highest levels of procollagen: 0x (A–C) and 5x SNZR PL (D–F). Days 18 and 20 are points described in Fig. 6 for procollagen production and day 20 was used in Fig. 5 to show the procollagen bands on a SDS gel stained with Sypro Ruby. In the early phases of this experiment the cells were grown without any serum but they initially seeded at low levels and on day 7 were still at low cell density. On day 12, 0.2% FBS was added to all the cell culture conditions in order to stimulate better proliferation. This was successful and on day 16 half the cultures were induced to make procollagen by adding ascorbate. The adult cells on day 20 seem to be less uniform then confluent embryonic cells (Fig. 4) but the older cells produce similar levels of procollagen when grown to high cell density and induced with ascorbate (Fig. 6).

procollagen if they are allowed to grow to high cell density (Table 1). Their response to ascorbate is also very similar to the embryonic tendon cells. Their proliferation rates seem to be lower than embryonic cells (Fig. 7) but the initial isolation procedure would have to be improved to confirm this. The adult cells reached a lower confluent density even in the presence of 0.2% FBS and various levels of SNZR PL and the morphology of the cells was more diverse including cells that were flatter and larger than confluent embryonic cells (Figs. 7C and 7F to Fig. 4 day 11). But most important, assays of collagen production in the intact tissue were ~1% that is more representative of an average protein compared (*Schwarz, Farson & Bissell, 1979*) to the high cell density cultures making large amounts. This is one the rare cases where cells in culture are more differentiated than *in vivo*.

## DISCUSSION

This article is focused on tendon cells, the signals they produce to regulate proliferation and differentiation, and the role played by the cell culture environment. In simple terms this article shows that a synthetic form of the chick tendon cell density signal can be used in the ng/ml range to drive tendon cells to proliferate. In addition, adult tendon cells can make prodigious amounts of procollagen similar to embryonic cells when grown in cell culture to high cell density and induced with ascorbate.

Examining the first point in more detail, this is confirmation of earlier observations that cell density signaling in tendons is due to a complex between a small, highly conserved protein and a unique, tissue-specific lipid (*Petzold & Schwarz, 2013*). Tissue-specific signals to control proliferation and differentiation seem essential in the embryo where different tissues are in close proximity to each other and crosstalk would be detrimental. With unique cell density signals, a tendon cell knows how many tendon cells are in its vicinity as well as how close they are to the muscle connective tissue (*Petzold & Schwarz, 2013*). This information allows the tendon cells to form a growth plate that can lay down an even distribution of collagen fibrils and at the same time control the length of the tendon so that there is no slack between the muscle and the bone at every stage in development from newborn to full-size adult. Dynamic engineering is a hallmark of structural tissues.

We should emphasize that our focus has been on the signaling method used by a tendon cell to sense the cell density in its vicinity. The finding that the signal has the characteristics of a short range diffusible factor has also been seen in U2OS cells (human osteosarcoma cells). Whether this is a more universal strategy or whether cell-cell membrane interaction is used by other cell types remains to be determined. Furthermore, most studies on cell density signaling have focused on signal transduction that control cell proliferation (*Fagotto & Gumbiner, 1996*; *Lieberman & Glaser, 1981*; *Maehama et al., 2021*; *McClatchey & Yap, 2012*; *Ribatti, 2017*; *Sharif & Wellstein, 2015*). Since we have only studied the initial step in cell density signaling in tendon cells, the level of overlap with other studies is not known. Similarly, the steps that link changes in the plasma membrane to an increase in procollagen production, increases in the endoplasmic reticulum, and many other cell density changes need to be defined.

Another critical aspect of cell density signaling in a tendon is feedback control using two factors. In developmental biology there is the common observation that cells proliferate to a high cell density and then growth slows while differentiation accelerates (*Kaldis & Richardson, 2012*). In a tendon, this is modified to form a growth plate that can lay down an even distribution of collagen fibrils while advancing in sync with the growth of the organism. What is not so clear is that the formation of a growth plate is driven by a two-factor model and their interactions with each other and the cell. Putting cells in culture does not change how the cell perceives its environment and its attempts to reestablish a growth plate (*Schwarz, Farson & Bissell, 1979*). SNZR PL binds to the outside of the plasma membrane and stimulates proliferation but as levels rise due to more tendon cells in the vicinity this also stimulates higher production of a non-diffusible factor that we postulate to be free SNZR L inside the cell. When the amount of SNZR L exceeds the amount of SNZR P, the excess is postulated to bind to the cytoplasmic side of the plasma membrane and interacts with SNZR PL on the outside part of the membrane. This complex becomes an inhibitor of proliferation and a stimulator of procollagen production. Free SNZR L also causes the cell to make less SNZR P. When the ratio of SNZR PL on the outside to SNZR L on the inside becomes small, the cell apoptoses. So this two-factor model gives the cell a sense of position, timing, and function (*Petzold & Schwarz, 2013*). We should emphasize the second factor is unknown. Free SNZR L is a likely candidate

because it is association with SNZR P on the outside of the cell and because it could bind to the cytoplasmic side of the plasma membrane. But we know that a mathematical model based on the characteristics we attribute to the second factor gives a remarkably accurate simulation of tendon morphogenesis (Fig. S1). The mathematical modeling was actually done before we knew the composition of SNZR PL, again using only its properties.

By adding SNZR PL to the medium, this alters the cellular response but in the unnatural manner caused by flooding the environment with SNZR PL. As a consequence, the levels have to be low in order to keep the cell from increasing the production of SNZR L and inhibiting proliferation. The optimum concentration was about 0.5 ng SNZR PL/ml. The cells would grow to a moderately high cell density and then 1 ng SNZR PL/ml pushed them to a confluent density. A confluent cell density is really an artifact of cell culture in that the cell is responding to other cells growing in a two-dimensional surface or the area of a circle. Cells *in vivo* are in a three-dimensional space and they respond to cells in a sphere of the same radius (~1 mm). So cells in the embryonic tendon when sectioned for histology on a two-dimensional slide look like cells at only a moderately high cell density but they produce high levels of procollagen and grow with a 48 h generation time (Fig. S1). So the environment can have a strong impact on how a short range signal is interpreted by the cell.

The cell culture environment also has a strong effect when the embryonic cells are first seeded into a cell culture flask or dish. The cells are coming from an *in vivo* state where there is a balanced level of SNZR PL and SNZR L that allows for high procollagen production and a fairly high growth rate. However, when you put them at low cell density in the presence of 10 ml of medium, the cells lose more SNZR PL as it diffuses into the large amount of medium. If the cell can replace some of that loss from its own production and from the production of the cells around them, then procollagen production will drop, and proliferation will stall but then comes back. If one tries to seed fewer cells, the distribution of cells over the dish becomes uneven. This problem becomes more apparent when the media to cell ratio is even greater. A good example is when cells are seeded into a 6 mm cloning ring at the same cell density as we seed cells across the whole dish (*Schwarz, 1991*). After the cells attach, the cloning ring is removed and the cells are exposed to 100x more media/cell. If too few cells are seeded in this condition, all the cells in the island apoptose within a few hours. What we see in this article, is that we can seed cells at lower densities with addition of SNZR PL at higher levels. The success is only temporary since higher levels of SNZR PL triggers an increase in SNZR L inside the cell that inhibits proliferation and these changes are not reversible. Basically, flooding the medium with SNZR PL is a poor delivery system for a short range signal at a local level. In some ways adding low levels of serum are better. Low serum levels are only sufficient for stimulating the cell to produce a slightly higher level of SNZR PL. The positive is that it maintains local delivery of the signal and the higher level allows a lower seeding density. The higher production of SNZR PL means that cells can grow faster resulting in the cells reaching confluent densities. The negative is that everything is speeded up and while we can postulate how serum works, it is difficult to prove. Nevertheless, it leads to the possibility of an improved delivery system for SNZR PL.

Can a local delivery system be designed for SNZR PL? For instance, one could design a protein with an affinity for the negatively charged cell culture plastic. The protein could have multiple fatty acids bound to it. The size of the lipid tail would determine how strongly it could interact with and bind to SNZR PL. If the number of particles and the binding strength was optimized, an embryonic tendon cell would sense being at moderately high cell density and would grow and make high levels of procollagen for a week. Since the level of SNZR PL would only get moderately elevated, the cells could be subcultured and the process repeated to yield an endless supply highly differentiated cells.

The second part of this article focused on adult tendon cells and whether in the presence of SNZR PL they would produce significant levels of procollagen. In adult tendon tissue the cells made low levels of collagen (~1%) (*Schwarz, Farson & Bissell, 1979*) and the cell density in the tissue is low. In major breaks in tendons or ligaments surgeon replace the tendons or ligament with either tendons or ligaments from other areas of the body or from cadavers. Minor tears can heal on their own but slowly and participation from tendon cells in the healing process is unclear (*Wu, Nerlich & Docheva, 2017*). Once again, putting adult tendon cells in culture and growing them to high cell density is not a normal situation for these cells. They remain responsive to ascorbate induction and to cell density. This actually solves one of the remaining puzzles of tendon morphogenesis. The growth plate explains how the tendon is always the right size by filling in the gap between the end of the tendon and the muscle. But the other problem is that the nascent tendon fibrils need to grow in diameter to support the weight of the growing chicken. There are cells at the perimeter of the growth plate that never reach high cell density and do not apoptose. These tendon cells are between the fibrils and in a perfect location to produce procollagen to expand the diameter of the fibrils. Again, cell density can be the critical regulator except in this setting it is inhibiting procollagen production as cell number declines, in line with a slowing down in the growth of the whole organism.

Understanding tendon morphogenesis, a tissue dominated by one cell type that is distinguished by its ability to make one protein in prodigious quantities, is still a complex problem. By showing that in tendon cell density signaling is driven by lipid\protein complex (SNZR PL) that can be synthesized, lifts the discussion beyond words to studies of how a two- factor model controls the cell. With a better delivery system for adding SNZR PL to cells in culture, this may allow primary cells to thrive at low seeding densities and for these cells to proliferate and make high levels of procollagen without being pushed to apoptose. Similar to the way embryonic tendon grows between embryonic days 12 through 18. This would allow tendon cell cultures to be subcultured and maintain growth and differentiation potential. The experiments with adult tendon tissue are encouraging in that they show that adult cells can produce prodigious amounts of procollagen when stimulated to grow to higher cell density. Potentially using SNZR PL and a better delivery system, regeneration of the tissue after a tear or break could be dramatically improved.

## ACKNOWLEDGEMENTS

Shengrong Li and his group at Avanti Polar Lipids for discussions about SNZR L and for the synthesis of the "alternative" structure. It was a pleasure to work with lipid experts that led to a successful conclusion.

Mina Bissell for her constant enthusiasm for good cell biology and for members of her lab who aided me in this research.

Jian-Hua Mao for constructive comments for improving the manuscript, encouragement and advice.

### Funding

The authors received no funding for this work.

### Competing Interests

I am the founder of SNZR LLC, a small company that is working on the commercial aspects of healing tendons and ligaments after injury.

The lipid/protein factor is covered by a patent: 10,428,093.

Title: Lipid cofactor essential for cell density signaling.

The company hired Avanti Polar Lipids to synthesize the unique lipid used in this article. So far, the company has a deficit of $50k.

### Author Contributions

- Richard I. Schwarz conceived and designed the experiments, performed the experiments, analyzed the data, prepared figures and/or tables, authored or reviewed drafts of the article, and approved the final draft.

### Patent Disclosures

The following patent dependencies were disclosed by the authors:

Patent #10,428,093 US date 10/1/2019.

Title: Lipid cofactor essential for cell density signaling.

### Data Availability

The raw data is available in the Supplemental Files.

### Supplemental Information

Supplemental information for this article can be found online at http://dx.doi.org/10.7717/peerj.14533#supplemental-information.

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
