# Peer review of "A synthetic cell density signal can drive proliferation in chick embryonic tendon cells and tendon cells from a full size rooster can produce high levels of procollagen in cell culture"

_PeerJ, doi:10.7717/peerj.14533_

## Round 0.1 · original submission · Major Revisions

The reviewers have identified several areas where improvement can be made to the manuscript, and accordingly this is classified as major revisions. We look forward to your response.

·

Basic reporting

In this well written manuscript, Dr. Richard I Schwarz examined the cell density-dependent collagen synthesis by cultured tenocytes isolated from tendon of chicken embryos and rooster. Author’ s previous studies have identified the SNZR PL, a complex containing SNZR P (a 94 amino acids protein) and SNZR L (lipid), which can support tendon cells growth in culture in lieu of FCS (fetal calf serum) and may interact with free SNZR L to mediate the signaling for the upregulated collagen synthesis by the cultured cells at confluence. It is hypothesized that a bi-factor mechanism involving both SNZR PL and free SNZR L might account for the up regulated collagen synthesis by confluent cultured tendon cells, but experimental data presented fall short to unequivocally support such hypothesis. For example, evidence of cell density-dependent SNZR L increase and interaction of SNZR PL and free SNZR L were not demonstrated, nor the presence of other factors were excluded.

Experimental design

Recombinant SNZR P, a 94 a.a. protein derived from a precursor protein, was produced in E. coli transfected plasmid DNA encoding the protein. The SNZR P and SNZR L mixture was incubated at 50~60°C to obtain SNZR PL complex that was not further characterized, e.g., molar ratio of SNZR P to SNZR L of the SNZR LP complex, Electron Microscope images of SNZR LP, the SNZR LP complex cannot be labeled by monoclonal anti-SNZR P antibodies (Maybe polyclonal antibodies recognize multiple epitopes in the 94 a.a. protein will be useful to recognize the SNZR PL complex). It remains elusive how SNZR P interact with SNZR L in cellulo to form SNZR LP complex. Does the precursor protein secreted into the culture medium or the conjugation of SNZR PL complex takes place inside cells?

Addition of 0.2% FCS is necessary to assist SNZR LP complex in culture medium to promote cell growth to confluency in timely fashion and up regulation of procollagen synthesis by both cultured embryonic and adult rooster tendon cells. The observation implicates all sides cell-cell interaction may have pivotal roles in signaling the cessation of proliferation and initiation of differentiation and gene expression patterns, respectively.

No data was available to elucidate whether SNZR LP did indeed interact with free SNZR L, rather than change of other cellular functions, e.g., altered expression of cytokines, which may contribute to the upregulation of collagen synthesis by cells in confluency.

Validity of the findings

Recombinant SNZR PL complex does indeed promote cells growth of tenocytes isolated from embryonic and adult rooster tendon. Both embryonic and adult rooster tendon cells display cell density-dependent collagen synthesis. It remains elusive whether the interaction of SNZR PL and free SNZR L accounts for up the up regulated synthesis by confluent tendon cells in culture. The data available does not support there is direct interaction between cell surface SNZR LP and intracellular free SNZR L. Other factors and/or changes of cellular gene expression patterns should also be considered.

Supplement Table 1 should be presented as Table 1, which is important to demonstrate that tendon cells of adult rooster tendon also express hi level of collagen synthesis.

Additional comments

Cell density-dependent collagen synthesis is an old observation of cultured tendon fibroblasts, which is probably true to most fibroblasts isolated from any connective tissues, if not all. However, the molecular and cellular mechanisms regulating collagen synthesis remains unknown. The author has made significant contribution in finding SNZR PL complex in promoting tendon fibroblasts and novel hypothesis of the interaction SNZR LP and free SNZR L may contribute to the regulation of cell density synthesis of collagen by fibroblasts. It would be of interest to determine whether all around and full cell-cell contact at confluency is perquisite for the upregulation of collagen synthesis or the cessation of cell growth, e.g., arrest of cell cycle, is sufficient for the up regulation of collagen synthesis.

Reviewer 2 ·

Basic reporting

In the manuscript titled "A synthetic cell density signal can drive proliferation 1 in chick embryonic tendon cells and tendon cells from a full size rooster can produce high levels of procollagen in cell culture, Dr Schwarz describes finding relating to the treatment of tendon cells with a lipid-protein complex that regulates their proliferation. I have a major and minor issue with the style of reporting.

The major issue: The manuscript occasionally reads as a review and other times as an original report. The abstract reflects the same: one is left confused as to where the previous reportage of relevant published observations end and where the new findings of this paper begin. A formidable deal of editing is sorely required before the manuscript can be accepted into the public domain. To illustrate with an example:
"They might
203 have to add both at the same time. And even then, this molecule might not be stable. A group of
204 lipid chemists and ms specialists at Avanti Polar Lipids met and discussed this problem. They
205 came up with an ìalternativeî structure (Fig. 1) where the fatty acids are reduced yielding an
206 alcohol with a ketone, a common metabolite."

It is not clear what was the process by which the alternative structure was homed upon. If its a hypothesis, then it shouldn't be part of results. If on the other hand it has been arrived through some prediction algorithm or through some thermodynamic minimization, that should be reported. The findings of Petzold & Schwarz, 2013 need to be carefully demarcated from this manuscript.

The minor issue is that there are numerous typographical and grammatical errors:
One example in the abstract: this allows the separation [of] the 66.8 kD protein from the vast majority of E. coli proteins.

Experimental design

I have one comment that is relevant to both the experimental design and the validity of findings section:

The SYPRO Ruby is an indirect way of ascertaining the fact that the lipid and protein have bound (Figure 2). Is there a more direct way of showing the same? Through some gel shift assay?

Validity of the findings

In case of the proliferation data, the cell numbers should be quantified. The current data in the manuscript is qualitative. Changes in cell number should be reflected through statistically tested observations over a few independent repeats of the experiment. If not possible, this should be discussed by the author.

---

## Round 0.2 · Minor Revisions

The manuscript corrections have been reviewed and are viewed as adequate and the paper improved, with only a minor adjustment needed in line 194.

·

Basic reporting

The revised manuscript has been greatly improved.

Minor revision is needed for the description of recombinant 94AA protein in line194 shoule be revised as "Because the cDNA encoding 94 AA protein contains Sap1 site----------"

Experimental design

Acceptable

Validity of the findings

Data presented support the hypothesis proposed.

---

## Round 0.3 · accepted · Accept

I am satisfied with the minor revisions to the manuscript; the paper is ready for publication.